# Validation of a Cell-Based Assay for Detection of Active Shiga Toxins Produced by *Escherichia coli* in Water

**DOI:** 10.3390/ijerph17217901

**Published:** 2020-10-28

**Authors:** Anna C. Hughes, Stephanie Patfield, Reuven Rasooly, Xiaohua He

**Affiliations:** 1Western Regional Research Center, United States Department of Agriculture, Agricultural Research Service, 800 Buchanan St., Albany, CA 94710, USA; anna.hughes@usda.gov (A.C.H.); steph@patfield.com (S.P.); reuven.rasooly@usda.gov (R.R.); 2Forensic Services Division, Contra Costa County Office of the Sheriff, 651 Pine St., Martinez, CA 94553, USA

**Keywords:** STEC, *E. coli*, waterborne, cytotoxicity, Shiga toxin, chlorine, HeLa cell

## Abstract

Shiga toxin-producing *Escherichia coli* (STEC) causes a wide spectrum of diseases, including hemorrhagic colitis and hemolytic uremic syndrome (HUS). Almost 5% of STEC infections result from waterborne exposures, yet there is no test listed in the EPA’s current Selected Analytical Methods for the detection of active Shiga toxins (Stxs) in water. In this study, a HeLa cell-based assay is validated for the detection of metabolically active Stxs produced by STEC in water, including tap, bottled, and pond water. Active Stxs are detected even when the number of Stx-producing bacteria is less than 0.4 CFU/mL and the assay performance is not affected by background flora or chlorine in the water. This assay is not only as simple and affordable as cell-free assays but also detects active holotoxins without the use of live animals. In addition, the assay is designed for use in multi-well formats, making it ideal for high-throughput screening of water samples and therefore useful for environmental public health surveillance programs to reduce human risk of infection with STEC.

## 1. Introduction

From 2009 to 2017, 603 reported Shiga-toxin-producing *Escherichia coli* (STEC) outbreaks resulted in 7869 illnesses and 1404 hospitalizations [1]. Roughly 4.5% of the reported STEC outbreaks during the 8-year period were a result of exposure to contaminated water [1]. There are many possible sources of waterborne STEC infections. Water mains can become exposed to STEC from damp soils near the site of repair or installation, whereas well water and aquifers can be contaminated by animal feces or sewage seepage [2,3,4,5,6]. Similarly, STEC can be found in recreational waters due to runoff or improper chlorination of pools [7,8]. Even bottled water is a potential source of infection, as O157:H7 has been shown to survive for up to 63 days in commercial bottled mineral water [9]. Nine (±5) waterborne STEC outbreaks are reported annually in the United States; however, it is thought that as few as 10% of the actual cases are reported and investigated. The actual number of outbreaks may be as much as 90% higher due to underreporting, detection limitations, and difficulties in tracking waterborne contaminants [1,2]. Thus, continued surveillance of *E. coli* contamination in drinking water is imperative for outbreak prevention.

STEC is capable of expressing a wide array of virulence factors, but Shiga toxins (Stxs) are the primary factor responsible for the development of severe complications like hemorrhagic colitis and hemolytic uremia syndrome [10,11,12]. Stxs are ribosomal inactivating proteins that can be divided into two serologically distinct groups: Stx1 (subtypes: Stx1a, Stx1c, Stx1d, and Stx1e) and Stx2 (Stx2 a–i, and Stx2k) [13,14,15,16,17,18,19,20]. STEC expressing Stx1a, Stx2a, Stx2c, or Stx2d are more often associated with human disease, but illnesses due to STEC expressing less common subtypes have been reported [21,22,23,24]. Stx subtypes vary in stability under extreme conditions and some can retain activity under conditions lethal to the organism, suggesting that consumption of products contaminated with active Stxs could result in illness even if the bacteria have been neutralized [25,26,27]. Because an effective therapeutic to treat *E. coli* infections does not exist, frequent water quality monitoring and continued improvement of STEC diagnostics are crucial.

Table 1 highlights a selection of methods for the detection of Shiga toxins. Each method has its advantages and disadvantages. In general, however, there is a tradeoff between the ability of an assay to detect toxin activity and the ability to differentiate Stx (sub)types. The methods recommended by the Environmental Protection Agency (EPA) in the Selected Analytical Methods (SAM) [28] for analyzing environmental water samples for Stxs can discriminate Stx types (ELISA) [29] and subtypes (LC-MS) [30] but cannot discriminate between active and inactive toxins [28]. The most common approaches that detect Stxs activity are mouse bioassays and cell-free assays [31,32]. Mouse bioassays are the gold standard for detection of active toxins but they can be technically challenging, require specific expertise and facilities, and are time-consuming. The cell-free assays are easier, fast, and cheaper but only detect the enzymatic activity of the A-subunit of Stxs. They do not monitor the activity of the holotoxins [27,33,34]. In contrast, cell-based cytotoxicity assays combine the benefits of the above two methods: they are rapid, inexpensive, and easy and detect the activity of the holotoxin.

Here, we demonstrate a cell-based assay that is capable of detecting active Stxs in water samples spiked with as few as 0.4 colony-forming units (CFU)/mL of STEC. This cell-based assay eliminates the need for using live animals, allows high-throughput screening, and yields results in a short amount of time. In addition, the performance of the assay is not affected by either the presence of chlorine at the concentrations used to treat drinking water or the background flora present in environmental samples.

## 2. Materials and Methods

### 2.1. Water Sample Collection

Samples of bottled water, tap water, and pond water were acquired from local sources in Alameda County, California, and collected in sterile tubes. All samples were collected in duplicate at 2 separate times. Alameda County treats tap water with chlorine, ammonia, fluoride, and sodium hydroxide [36].

### 2.2. Stx Enrichment and CFU Determination

All strains used are listed in Table 2.

Single colonies of *E. coli* strains with or without *stx1a*, *stx2a*, or both *stx1a* and *stx2a* genes were inoculated into tryptone soya broth (TSB) and grown at 37 °C for 18 h. The overnight culture was then serially diluted into Bacto peptone water (BPW) to 10 CFU/mL. Actual inoculum levels were later determined by spread-plating 0.1 mL of the cultures onto tryptone soya agar (TSA) plates and incubating them overnight at 37 °C for manual colony counting. One mL of diluted culture in BPW was added to 24 mL of the water samples and enriched with 75 mL TSB containing 50 ng/mL mitomycin C (MMC) and 10 g/L casamino acids, which was then grown at 42 °C for 16 h. A 25 mL water-only control (“un-inoculated”) was enriched using the same protocol as the inoculated samples. Following overnight enrichment, 500 μL of each sample was lysed with an equal volume of phosphate B-PER reagent (ThermoFisher Scientific, Waltham, MA, USA) for 1 h at 37 °C and centrifuged at 13,000× *g* for 10 min at 4 °C. Bacterial supernatants were collected and filtered through a 0.2 μm filter and used in cytotoxicity assays as described below (Figure 1).

### 2.3. Stx2a Protein Purification

Stx2a was purified from MMC-induced *E. coli* supernatants (RM10638) by affinity chromatography using an affinity column (Thermo Scientific AminoLink Plus Immobilization Kit, Waltham, MA, USA) coupled to mAb Stx2–1, which binds the A-subunit, and mAb Stx2–5, which binds both the A- and B-subunits, followed by gel filtration on an AKTA FPLC using a Superdex 200-XK 26/70 column (GE Healthcare, Marlborough, MA, USA) as described previously [40].

### 2.4. Cytotoxicity Assays

Stxs cytotoxicity was determined using a microtiter HeLa cell assay [38]. A 96-well plate (Corning) was seeded with 10^4^ Linterna tGFP HeLa cells (InnoProt, Spain) in high-glucose Dulbecco’s Modified Eagle Medium (DMEM) supplemented with 1× Glutamax, 250 µg/mL G418 and 10% fetal bovine serum (FBS). Plated HeLa cells were treated with the indicated Stx condition (see below) and incubated for 24 h at 37 °C and 5% CO_2_.

Following Stx treatment, cytotoxicity was assessed by the addition of Cell-Titer Glo (CTG, Promega, Madison, WI, USA) according to the manufacturer’s instructions, with the exception of diluting the reagent 1:5 in Phosphate-Buffered Saline (PBS) (Figure 1). Luminescence was measured for 0.1 s in a VictorIII plate reader (Perkin Elmer, Waltham, MA, USA). Luminescence measurements were used to calculate percent survival and percent cytotoxicity as described below. All conditions were assayed in triplicate and repeated twice.

#### 2.4.1. CD_50_

The concentration of Stx2a required to reach 50% of cell death (CD_50_) was determined by incubating HeLa cells with 10-fold serial dilutions of pure Stx2a from 10 fg/mL to 100 ng/mL in DMEM for 24 h at 37 °C with 5% CO_2_. Percent survival was calculated as: (counts per second (CPS) of the experimental sample/average CPS DMEM-only control) × 100% and plotted on a semi-log scale. The linear portion of the dose response curve was used to estimate the concentration of Stx2a needed to reach CD_50_.

#### 2.4.2. Water

To determine cytotoxicity of Stxs in water samples, 5 µL of the enriched samples was added to HeLa cells and incubated for 24 h at 37 °C with 5% CO_2_ in 96-well plates. The percent toxicity was calculated as (the average CPS of the DMEM-only control − the CPS of the experimental sample)/average DMEM control × 100%.

#### 2.4.3. Chlorine

The effect of chlorine on HeLa cell viability was assessed by preparing 2-fold dilutions of sodium hypochlorite (bleach) from 0.2–3.2 mg/L in sterile water. The diluted chlorine (5 µL) was then added to 10^4^ HeLa cells in DMEM and incubated for 24 h at 37 °C with 5% CO_2_.

## 3. Results and Discussion

The EPA’s current Selected Analytical Methods (SAM) recommended for Stx detection in environmental samples do not include a method that differentiates between active and inactive toxins. Therefore, we wanted to determine if a cell-based assay is appropriate for detection of active Stxs present in environmental water samples. First, we tested the cytotoxic effect of Stx2a on the human cell line available in our lab. A dose response was observed when Stx2a was diluted 10-fold between 100 fg/mL and 100 pg/mL in a 96-well microtiter plate seeded with 10^4^ HeLa cells. A CD_50_ (the toxin concentration that kills 50% of cells) of 23.38 pg/mL ± 0.6 was observed (Figure 2). To confirm that Stx2a was responsible for the cytotoxicity, a neutralization test was performed by pre-incubating Stxs with Stx2-specific mAb prior to incubation in the human cells. Consistent with previously reported results, the addition of Stx2a mAb neutralized the cytotoxicity of Stx2a to the HeLa cells (data not shown), confirming the specificity of the assay [38]. We note that the CD_50_ reported here is lower than previously reported and thus appears to be more sensitive to Stx2a [38]. The difference is likely due to the use of different cell lines (GFP expression) or slight differences in toxin preparation due to inter-researcher variability and further highlights how comparisons of absolute toxicity across studies are not valid [25]. Nevertheless, we reconfirm that HeLa cells can be used to detect pure active Stx and emphasize the importance of standardized controls within and between experiments.

Cell culture assays are notoriously sensitive to changes in growth conditions and there are various components present in different water sources with the potential to introduce assay variability [41,42,43,44]. Therefore, various water sources (bottled, tap, and pond water) were tested for their effect on HeLa cell growth. First, the background flora present in samples was determined via total aerobic plate counting. No microbial growth was detected in bottled or tap water samples. The background level detected in pond water was 5 × 10^2^ CFU/mL but did not appear to affect the HeLa cell assay compared with the bottled and tap water samples (Table 3). Moreover, none of the drinking water sources had an effect on HeLa cell viability compared with the negative control (Table 3).

The EPA allows up to 4 mg/L of chlorine (NaOCl) as a drinking water disinfectant; however, it was unknown if HeLa cell performance is affected by chlorine [45]. To assess HeLa cell viability in the presence of the chlorine specifically, the disinfectant was serially two-fold diluted into pure water from 0 to 3.2 mg/L and 5 µL was added to 10^4^ HeLa cells. After an overnight incubation with the chlorinated water, HeLa cells remained viable across all concentrations tested (Figure 3). This suggests that HeLa cell performance would not be affected by the presence of chlorine when used to assay for biologically active Stxs in chlorinated water.

STEC infections are characterized by an extremely low human infectious dose: fewer than 10 cells are sufficient to cause illness [46]. To validate the capability of the HeLa cell assay for the detection of live STEC present in environmental water samples using Stx as a marker, water samples, including bottled, tap, and pond water, were spiked with a low infectious dose (6–12 CFU/25 mL) of *stx*-positive and negative strains. Table 3 demonstrates that water samples inoculated with STEC producing Stx1a, Stx2a, or both Stx1a/2a exhibited 63–76% cytotoxicity, whereas less than 1% toxicity was detected in un-inoculated samples or samples inoculated with a non-STEC strain. No significant differences (*p* > 0.05) in cytotoxicity were detected in samples (bottled, tap, and pond water) inoculated with the same STEC strains (Table 3), suggesting there were no matrix effects associated with these samples. These results indicate that the HeLa cell assay could detect STEC in water unambiguously. Relative to other activity assays, this assay is simple, sensitive, and reliable and would complement the current Stx assays recommended by the EPA.

## 4. Conclusions

The Stx enrichment and HeLa cell cytotoxicity assay developed here detects enzymatically active holo-Stxs produced at less than 0.4 CFU/mL of STEC in water. Common water additives, such as chlorine and fluoride, or background flora present in water do not appear to interfere with the assay or cause any growth competition for the relatively low inoculum levels of STEC targeted. This HeLa cell cytotoxicity assay holds promise for the identification of live STEC present in environmental water by food industries, the EPA, or other regulators.

## Figures and Tables

**Figure 1 ijerph-17-07901-f001:**
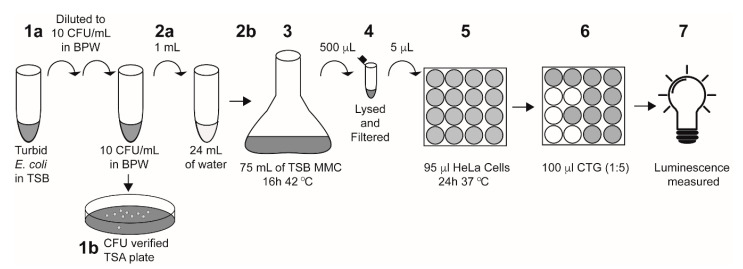
A schematic of the *E. coli*-inoculated water sample preparation and HeLa cell assay. Step 1a: a turbid overnight culture was serially diluted into BPW to 10 CFU/mL. Step 1b: 100 μL of the diluted culture was plated onto TSA plates to verify inoculum levels. Step 2a: 1 mL of the diluted *E. coli* culture was added to 24 mL of the indicated water sample. Step 2b: the entire 25 mL water–*E. coli* mixture was added to 75 mL of TSB containing 50 ng/mL mitomycin C and 10 g/L casamino acids. Note: for un-inoculated controls, 25 mL of the indicated water sample was added to the TSB without the 1 mL of *E. coli*. Step 3: Culture was enriched at 42 °C for 16 h. Step 4: 500 μL of the enriched sample was lysed with the addition of 500 μL of B-PER reagent, then filter-sterilized. Step 5: The enriched filtered lysate (5 μL) was added directly to 95 μL of HeLa cells in a 96-well plate and incubated at 37 °C for 24 h with 5% CO_2_. Step 6: 100 μL of Cell-Titer Glo (CTG) reagent was added to each well. Step 7: Luminescence was measured on a plate reader.

**Figure 2 ijerph-17-07901-f002:**
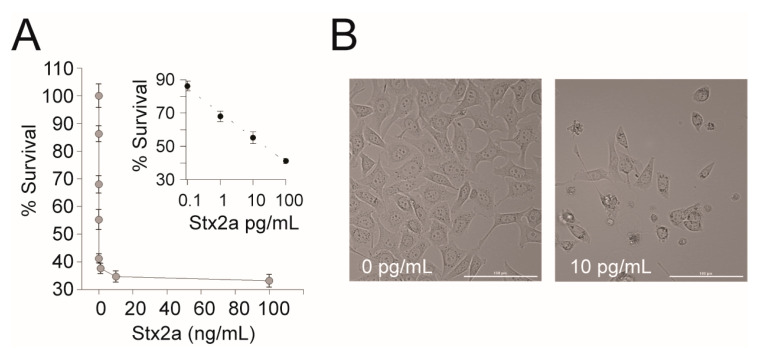
HeLa cell assay for Stx2a activity. (**A**) A representative dose response curve of Stx2a-treated HeLa cells. HeLa cells were treated with 10-fold serial dilutions of Stx2a from 10 fg/mL to 100 ng/mL in DMEM for 24 h at 37 °C with 5% CO_2_. Cell viability was measured using Cell Titer Glo and luminescence was measured on a Victor3 plate reader. Percent survival was calculated as (CPS of the experimental well/CPS of the DMEM only control) × 100%. Inset: The linear portion of the semi-log transformed curve. The dashed line is the best-fit equation used to calculate the CD_50_ (*y* = −6.425 ln(*x*) + 70), *R*^2^ = 0.99. Error bars represent the standard deviation of the mean from three replicates. (**B**) Representative images (20× magnification) of HeLa cells intoxicated with Stx2a or a medium-only control. Scale bar is 100 µm.

**Figure 3 ijerph-17-07901-f003:**
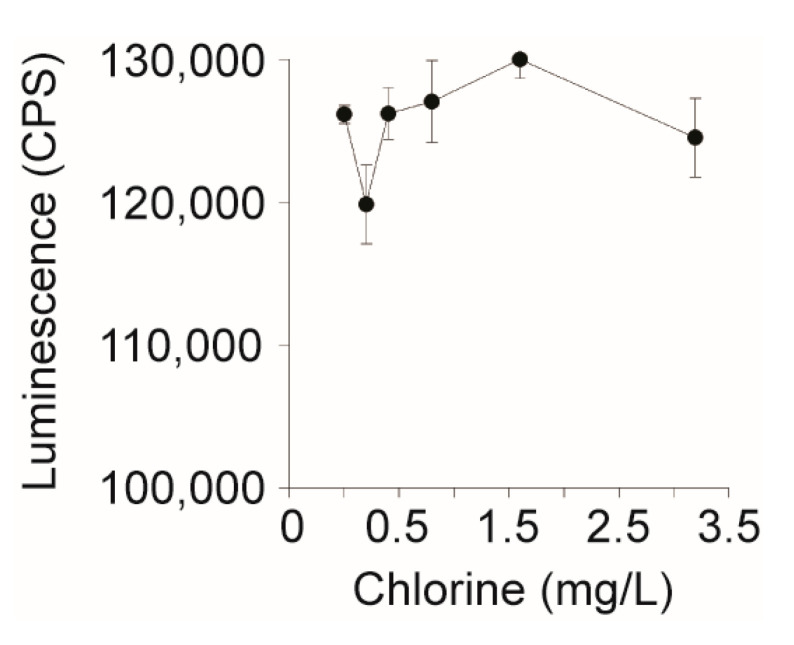
Effect of chlorine levels on the performance of the HeLa cell-based assay. Sodium hypochlorite (bleach) was diluted two-fold from 3.2 to 0.2 mg/L and 5 µL was added to 10^4^ HeLa cells in DMEM, which were incubated for 24 h at 37 °C with CO_2_. Cell viability was measured using Cell Titer Glo and luminescence was measured on a Victor3 plate reader.

**Table 1 ijerph-17-07901-t001:** Common tests used for Stxs.

Assay Type	Assay Time	LOD ^a^	Detection Principle	Detects Activity
ELISA [29]	2.5 h	25 pg/mL	mAb ^b^ capture, mAb detection	No
LC-MS [30]	5–6 h	5 fmol/mL	Conserved peptides (tryptic digestion of the B subunits)	No
Mouse bioassay [35]	5 days	290 ng/kg	Uses death rate to estimate LD50 ^c^	Yes
Cell-free translation [33]	90 min	20 pg/mL	Luciferase protein synthesis	Yes
HeLa assay *	2 days	23 pg/mL	ATP assay for detection of viable cells	Yes

* This study. ^a^ LOD, limit of detection; ^b^ Monoclonal Antibody; ^c^ Lethal Dose, 50%.

**Table 2 ijerph-17-07901-t002:** Strains used in this study.

Organism	Strain	Stx Expressed	Reference/Collection
*E. coli*	RM13506	Stx1a	[37]
*E. coli*	RM10638	Stx2a	[38]
*E. coli*	RM7370	Stx1a and Stx2a	[39]
*E. coli*	ATCC25922	n/a	ATCC^®^ 25922™
Human	tGFP-HeLa (LINTERNA)	n/a	P20107

**Table 3 ijerph-17-07901-t003:** Detection of Stx cytotoxicity in water samples inoculated with bacteria.

Bacterial Strain *	No. Sample Control	Bottled Water	Tap Water	Pond Water
Un-inoculated	106,790 ± 2319 (0%)	112,485 ± 4136 (−5%)	111,440 ± 509 (−4%)	119,550 ± 1145 (−12%)
RM13506-Stx1a		25,190 ± 791 (76%)	28,230 ± 197 (74%)	28,310 ± 1357 (73%)
RM10638-Stx2a		35,900 ± 9687 (66%)	34,895 ± 431 (67%)	32,710 ± 395 (69%)
RM7370-Stx1a/2a		30,945 ± 2962 (71%)	39,810 ± 565 (63%)	36,735 ± 2849 (66%)
ATCC25922-no Stx		106,225 ± 926 (1%)	111,035 ± 487 (−4%)	112,725 ± 3146 (−6%)

Data represent counts per second ± standard deviation (cytotoxic dose). * The predicted inoculum level of each strain was 0.4 CFU/mL; the actual inoculum levels ranged from 0.24 to 0.48 CFU/mL.

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
