# Peer review of "Validation of a Cell-Based Assay for Detection of Active Shiga Toxins Produced by Escherichia coli in Water"

_ijerph, 2020, doi:10.3390/ijerph17217901_

Round 1

Reviewer 1 Report

Due to high lethal toxicity of Shiga toxins it is important to have rapid and sensitive methods to detect these toxin produced by some Escherichia coli strains in waters.  

This paper would get more readers if it would be clearer.

Title: Omit environmental!

Abstract: Bacteria are measured as colony forming units in mL and toxin as ng (or other weight unit)  in mL. 

Line 16 would be better as … active Stxs could be detected even when the number of Stxs producing bacteria was less than 0.4 CFU/mL.

Material and methods

2.1. You inoculated waters with Stxs toxin producing bacteria. Inform this!  

Line 74: MMC? Is it mitomycin  C? If, add!

Line 82: 100 L or 100 µL?

Line 83: You used CellTiter Glo luminescent cell viability (or survival rate or cytotoxicity)  assay and luminescence was measured as counts per second?  It is partly opened in caption of  Fig 1. Explain better near the line 83!

Line 90 CD50: as well as WATER: (line 96) and CHLORINE: (line 100)  

The 4 ppm Cl is high, but if you wanted to study the analysis of shiga toxin, so this high concentration is ok.

How long it takes to get a result of Shiga toxins?

Line 96: is 5 L correct?

Results and Discussion

See the lines 101 and 141: p.p.m or ppm? .

Line 130: R2  (not R2)

Fig 1 A is clear, but Fig 1 B is not visible.  If you are not able to improve it, omit it totally.

Table 2: Do you want to show the difference between water or between bacteria strains inoculum levels?  If you wish to compare bacteria and inoculum levels, turn the table so that bacteria and their inocula form columns and waters forms  rows. Present  CPS±SD  values and (CD )!  Could the background flora be in footnote?  In this case the reader can easily compare the cytotoxicities of  different strains and with similar inoculum levels in different waters. I think that you would like to show that the method gave the same results in tap water, pond water and bottle water.

Think still the accuracy of your results!  Often the weakest point is the measuring of volume which can be 1.00 mL but not 1.000 mL or maybe 1.0 µL but hardly 1.00 µL.  Thus maybe 3 numbers are really trustful.

Correct all references. All names and if you refer a journal article follow:
1. Author 1, A.B.; Author 2, C.D. Title of the article. Abbreviated Journal Name YearVolume, page range.

Author Response

Response to Reviewer 1 Comments

Point 1: This paper would get more readers if it would be clearer.

Response 1: Thank you, we have attempted to clarify several points based on your, and other reviewers, suggestions.

Point 2: Title: Omit environmental!

Response 2: Omited

Point3:

Line 16 would be better as … active Stxs could be detected even when the number of Stxs producing bacteria was less than 0.4 CFU/mL.

Response 3: Thank you, line 16 has been revised and is now line 17.

Point 4: 2.1. You inoculated waters with Stxs toxin producing bacteria. Inform this!  

Response 4: Section 2.1 is only describing the collect of water samples. The inoculation process is described in section 2.2. However, we have added a figure schematic for further clarification. See Figure 1.

Point 5: Line 74: MMC? Is it mitomycin  C? If, add!

Response 5: It is mitomycin  C, and it has been corrected.

Point 6: Line 82: 100 L or 100 µL?

Response 6: Corrected to µL

Point 7: Line 83: You used CellTiter Glo luminescent cell viability (or survival rate or cytotoxicity)  assay and luminescence was measured as counts per second?  It is partly opened in caption of Fig 1. Explain better near the line 83!

Response 7: Sections 2.4.1 -2.4.3 are dedicated specifically to describing the analysis of luminescent data for each assay. We found it even more confusing when it was all in the same section. However, we have now added a line at 120-121 to make it clearer these sections are connected.

Point 8:Line 90 CD50: as well as WATER: (line 96) and CHLORINE: (line 100)  

Response 8: Colon added.

Point 9: The 4 ppm Cl is high, but if you wanted to study the analysis of shiga toxin, so this high concentration is ok.

Response 9: Yes, this is a great point. When we designed the experiment we did not think that chlorine in drinking water could be as high as 4 ppm. However, we did test tap water which, we do not know the exact concentration of chlorine present, but in combination with the many other additives, we presumed it would be the most likely condition to affect HeLa cell growth. We would like to add the experiments with the higher concentration of Cl specifically, but due to the pandemic and shelter in place orders we are unable to conduct any experiments currently.

Point 10: How long it takes to get a result of Shiga toxins?

Response 10: Two days using our assay. See Table 1.

Point 11: Line 96: is 5 L correct?

Response 11:  It should read “5 µL of enriched samples”

Point 12: See the lines 101 and 141: p.p.m or ppm? .

Response 12: Corrected to mg/L per reviewer 2 request.

Point 13: Line 130: R2  (not R2)

Response 13: Changed to R2 

Point 14: Fig 1 A is clear, but Fig 1 B is not visible.  If you are not able to improve it, omit it totally.

Response 14: Fig 1B has been revised.

Point 15: Table 2: Do you want to show the difference between water or between bacteria strains inoculum levels?  If you wish to compare bacteria and inoculum levels, turn the table so that bacteria and their inocula form columns and waters forms  rows. Present  CPS±SD  values and (CD )!  Could the background flora be in footnote?  In this case the reader can easily compare the cytotoxicities of  different strains and with similar inoculum levels in different waters. I think that you would like to show that the method gave the same results in tap water, pond water and bottle water.

Response 15: Thank you for the suggestion, we have updated the table.

Point 16: Think still the accuracy of your results!  Often the weakest point is the measuring of volume which can be 1.00 mL but not 1.000 mL or maybe 1.0 µL but hardly 1.00 µL.  Thus maybe 3 numbers are really trustful.

Response 16: I am sorry that I do not understand the nature of this comment. We kept all of our numbers to 1 decimal place or no decimal places.

Point: Correct all references.

Response : I apologize for the formatting errors. I was using the wrong endnote format. I hope they are correct now.

Reviewer 2 Report

In this manuscript, the authors demonstrated the development of a cell-based assay for the detection of Shiga toxin that produced by E. coli. This assay revealed high validation for the detection of active Stxs with a concentration less than 0.4 CFU/mL, and the detection was not affected by background flora or chlorine in water. Therefore, this simple cell-based assay exhibited great potential for highly sensitive detection of active holotoxins in water samples. It is a very interesting work. However, the authors should add more information and control experiments to make it to be a complete story. Therefore, a major revision is necessary.

Special comments:

  1. It is suggested for the authors to add a scheme to make it more clear for the cell-based assay.
  2. In Figure 1b, it is hard to see the difference in the three optical images. It is suggested for the authors to provide images with higher resolution. In addition, it will be better if there authors to try the CLSM characterization to obtain color images.
  3. In Figure 2, the error bars should be added to the data points.
  4. The selectivity and stability of the cell-based assay should be studied and discussed.
  5. It is necessary for the authors to evaluate the detection performances of Shiga toxins by comparing with other previously reported methods. It will prove the importance of this cell-based assay for detecting Shiga toxins.

Author Response

Response to Reviewer 2 Comments

Point 1: It is suggested for the authors to add a scheme to make it clearer for the cell-based assay.

Response 1: This is a great idea. Thank you for the suggestion. The scheme has been added and is now figure 1.

Point 2: In Figure 1b, it is hard to see the difference in the three optical images. It is suggested for the authors to provide images with higher resolution.

Response 2:The images in Figure 2 B have been modified and are hopefully more visible.

Point 3: In Figure 2, the error bars should be added to the data points.

Response 3: Error bars have been added.

Point 4: The selectivity and stability of the cell-based assay should be studied and discussed.

Response 4: This is a very good point. HeLa cells can be used to detect cytotoxicity of many toxins (Ricin, Abrin, Stx etc.) To address this, we have added a line to show specificity by using a mAb-stx2a to neutralize the effect of stx toxin on HeLa cells see line 144-148. However, we would like to note that an in depth analysis of stability and specificity of the HeLa cell assays has been published previously and we believe is beyond the intended scope of this Brief Report which is to demonstrate that this assay is appropriate for detection of Stx in water.

Point 5: It is necessary for the authors to evaluate the detection performances of Shiga toxins by comparing with other previously reported methods. It will prove the importance of this cell-based assay for detecting Shiga toxins. -

Response 5: We have added more information (table 1and Lines 50-62 ) comparing the cell based assay with the other assays discussed in this manuscript.

Reviewer 3 Report

Revision of the Manuscript ID: ijerph-971021 entitled “Validation of a cell-based assay for detection of active Shiga toxins produced by Escherichia coli in environmental water”

The manuscript addresses the application of a new, fast, and reliable tool to detected Shiga toxins produced by STECs. The topic has a potential applied interest and despite the study of STECs is wide, this is a novel study with a new detection technique.

However, the manuscript has small weaknesses, and I have some concerns prior accepting this manuscript to the International Journal of Environmental Research and Public Health. The authors must address the comments and or justify some putative limitations. In detail bellow.

a) Title: the title is adequate to the work and follows journal guidelines

b) Keywords: 7 keywords are used (journal standards are 3-10, so ok here); keywords are fine, they are specific to the work.

c) Abstract: abstract is well written and addresses the different parts of the work (introduction with objective, materials and methods, results and discussion, conclusion), which I find to be a good abstract.

d) Introduction: introduction is well written, addressing the state of the art of the problem suggested by the authors. The references used support the information described. However, I would like to see the introduction more developed (it is rather small, and too concise), in this part of the manuscript the authors should give more content about the state of the art of this problem.

e) Abbreviations: Ok

f) Materials and methods: This section is well written and well explained, in order that other researchers can replicate the methods if the case arrives. However small details need clarification.

2.1) how many samples were collected from each source? The authors should include this information in this subsection

2.2) Legend of Table 1 should include the species name of the strains used.

g) Results and Discussion: this section is well written, the results are presented in a clear way, and the images and the tables complement the text. In regards of the discussion, it is a good discussion, sufficing good information to compare, with suitable references, it is well written, and overall great. Once the results and discussion are together, the authors should be careful to not repeat the results too much when discussing them, a revision of the discussion part should be made to minimize these repetitions.

h) Conclusion: Concluding remarks are well written, however they repeat some information stated in the discussion, the authors should try to minimize these repetitions.

i) Figures and tables: overall ok.

j) References: references are adequate to the work, and the authors only use 5 self-citations, which I find a positive aspect of the manuscript. In the total of 44 citations, 23 are from the last ten years, which is a positive point.

Author Response

Response to Reviewer 3 Comments

Point 1: Introduction: introduction is well written, addressing the state of the art of the problem suggested by the authors. The references used support the information described. However, I would like to see the introduction more developed (it is rather small, and too concise), in this part of the manuscript the authors should give more content about the state of the art of this problem.

Response 1: The intro has been revised and expanded. Please see lines 47-56 and the new Table 1.

Point 2: How many samples were collected from each source? The authors should include this information in this subsection

Response 2: Water samples were collected twice , at different times, and tested in triplicate. This information was added to the manuscript. At lines 75 and 122.

Point 3: Legend of Table 1 should include the species name of the strains used.

Response 3: A column was added to indicate organism.

Point 4: Results and Discussion: this section is well written, the results are presented in a clear way, and the images and the tables complement the text. In regards of the discussion, it is a good discussion, sufficing good information to compare, with suitable references, it is well written, and overall great. Once the results and discussion are together, the authors should be careful to not repeat the results too much when discussing them, a revision of the discussion part should be made to minimize these repetitions.

Response 4: The discussion has been reworded to reduce repetitions

.

Point 5: Conclusion: Concluding remarks are well written, however they repeat some information stated in the discussion, the authors should try to minimize these repetitions.

Response 5: The conclusion has been rewritten, hopefully it’s less repetitive.

Reviewer 4 Report

The manuscript reports important work on the evaluation of a cell-based assay as complementary or even superior method to the ones already approved by US EPA for reliable detection of STEC in water samples. The manuscript fits the scope of the journal and should be published.

I only have a few editorial comments:

Abstract

Line 13, water

Line 18, ppm should be replaced by mg/L

Materials and Methods

Lines 59, 82, 96, and 102, I presume this shall mean microliter?

Line 74, MMC shall be explained

References

Please use journal abbreviations in all cases, please only indicate volumes but not issues

Author Response

Response to Reviewer 4 Comments

All requested edits were addressed. Thank you.

Round 2

Reviewer 2 Report

In this revised version, the authors made great improvement according to the comments and suggestions of all referees. Now it is suitable for publication at this journal.